# Cyclin-Dependent Kinase Inhibitors in the Rare Subtypes of Melanoma Therapy

**DOI:** 10.3390/molecules29225239

**Published:** 2024-11-06

**Authors:** Jonatan Kaszubski, Maciej Gagat, Alina Grzanka, Agata Wawrzyniak, Wiesława Niklińska, Magdalena Łapot, Agnieszka Żuryń

**Affiliations:** 1Vascular Biology Student Research Club, Department of Histology and Embryology, Faculty of Medicine, Collegium Medicum in Bydgoszcz, Nicolaus Copernicus University in Toruń, 85-067 Bydgoszcz, Poland; johnny.kaszubski@gmail.com; 2Department of Histology and Embryology, Faculty of Medicine, Collegium Medicum in Bydgoszcz, Nicolaus Copernicus University in Toruń, 85-067 Bydgoszcz, Poland; agrzanka@cm.umk.pl (A.G.); azuryn@cm.umk.pl (A.Ż.); 3Faculty of Medicine, Collegium Medicum, Mazovian Academy in Płock, 09-402 Płock, Poland; m.lapot@mazowiecka.edu.pl; 4Department of Histology and Embryology, Institute of Medical Sciences, College of Medical Sciences of the University of Rzeszow, University of Rzeszów, 35-310 Rzeszów, Poland; awawrzyniak@ur.edu.pl; 5Department of Histology and Embryology, Medical University of Bialystok, 15-269 Białystok, Poland; wieslawa.niklinska@umb.edu.pl

**Keywords:** melanoma, rare subtypes of melanoma, mucosal melanoma, uveal melanoma, acral melanoma, cyclin-dependent kinases (CDKs), CDK inhibitors (CDKIs), CDK4/6, CDK7/9, CDK2

## Abstract

Melanoma occurs in various forms and body areas, not only in the cutis, but also in mucous membranes and the uvea. Rarer subtypes of that cancer differ in genomic aberrations, which cause their minor sensibility to regular cutaneous melanoma therapies. Therefore, it is essential to discover new strategies for treating rare forms of melanoma. In recent years, interest in applying CDK inhibitors (CDKIs) in cancer therapy has grown, as they are able to arrest the cell cycle and inhibit cell proliferation. Current studies highlight selective CDK4/6 inhibitors, like palbociclib or abemaciclib, as a very promising therapeutic option, since they were accepted by the FDA for advanced breast cancer treatment. However, cells of every subtype of melanoma do not react to CDKIs the same way, which is partly because of the genetic differences between them. Herein, we discuss the past and current research relevant to targeting various CDKs in mucosal, uveal and acral melanomas. We also briefly describe the issue of amelanotic and desmoplastic types of melanoma and the need to do more research to discover cell cycle dysregulations, which cause the growth of the mentioned forms of cancer.

## 1. Introduction

Melanoma is considered the most dangerous and deadliest form of cutaneous cancer, although it constitutes only 4% of all skin cancer cases. It develops from melanocytes—cells located in the basal layer of the epidermis—which produce pigment melanin. Melanin is responsible for absorbing ultraviolet radiation (UV) and protecting the skin, but UV can cause DNA mutations, which lead to cancer [1].

Significantly, melanoma can attack not only the skin, but also mucous membranes and the uvea. For that reason, we can distinguish four subtypes of the disease: cutaneous, acral, mucosal and uveal melanoma [2]. According to another classification, the most common types of melanoma are superficial spreading, nodular and lentigo maligna melanoma. Mucosal, uveal, acral lentiginous, amelanotic, desmoplastic and polypoid melanoma are less common types, but we cannot underestimate them, as they still feature high mortality rates [3]. People of the Caucasian race have a higher risk of having cutaneous melanoma, which arises in nonglabrous skin. However, acral melanoma, which develops in the palms, soles and nails, and mucosal melanoma frequently attack Black, Asian or Hispanic people [1].

Traditional ways of cancer treatment, including surgical treatment or chemotherapy, do not always work in rarer forms or higher stages of melanoma. A discover of the mutational landscape led us to therapies targeted at specific mutations like BRAF (B-Raf proto-oncogene, serine/threonine kinase) and MEK (mitogen-activated protein kinase). BRAF/MEK inhibitors like vemurafenib, dabrafenib and trametinib are often used in the treatment of melanoma with BRAF V600 mutations [3,4], but those therapies also have their limitations [1,2]. Since the mid-1990s, there has been a growing interest in therapies that focus on inhibiting cell proliferation by targeting those proteins that control the cell cycle, which are cyclins and serine/threonine cyclin-dependent kinases (CDKs) [5]. Some of the novel CDK inhibitors (CDKIs), selectively targeting specific CDKs, have been already used in treating advanced breast cancer. Recent studies pointed to CDKIs as a promising option for melanoma treatment and potential candidates for synergistic therapies in combination with BRAF/MEK inhibitors [2,5].

In this review, we present the overall profiles of mucosal, uveal and acral melanoma and the cell cycle dysregulations, which cause their progress. We take a closer look at preclinical and clinical studies relevant to the use of cyclin-dependent kinase inhibitors (CDKIs) in the treatment of those rare melanoma subtypes. Eventually, we briefly discuss those types of melanoma, whose cell cycle aberrations are barely discovered, and suggest the future prospects of their treatment. In preparing this review, we searched the PubMed database using the following keywords: “uveal melanoma CDK4/6 inhibitors”, “uveal melanoma CDK4 inhibitors”, “uveal melanoma CDK2 inhibitors” and “uveal melanoma CDK7/9 inhibitors” as well as similar terms for other types of melanoma and targeted kinases. We focused particularly on preclinical trials relevant to uveal melanoma, as these have been scarcely mentioned in previous review articles. To our knowledge, desmoplastic and amelanotic forms of melanoma have also not been discussed in terms of the role of CDKs and CDK inhibitors, likely due to the limited amount of research in this area. For this reason, we decided to briefly address them at the end of the article.

## 2. The Role of CDK Inhibitors in the Cell Cycle

Eukaryotic cell proliferation occurs in the cell cycle that consists of the M phase and three stages of interphase: G1, S and G2. There is also a G0 stage, responsible for the resting of the cell, in case of its inability to proliferate. During the G1 phase, the cell is being prepared for DNA replication (S phase), whereas the G2 phase prepares the cell for the mitosis and cytokinesis that take place during the M phase. The cycle is regulated by a group of proteins including cyclins, serine/threonine cyclin-dependent kinases (CDKs) and cyclin-dependent kinase inhibitors (CDKIs) [6].

In the G1 phase, the complex of CDK4/6 and D-type cyclin (D1, D2 or D3) is implicated, which leads to phosphorylation of retinoblastoma protein (RB1). The E2 transcription factor family (E2F) starts releasing from the RB1-E2F complex and begins the synthesis of cyclin E. Cyclin E binds with CDK2 kinase and allows the cell to enter the S phase and start DNA replication, which is later controlled by the cyclin A–CDK2 complex. Next, the cyclin A–CDK1 and cyclin B–CDK1 complexes lead to S/G2 and G2/M transitions [6].

Except for cyclin-dependent kinases which regulate the cycle (CDK4, CDK6, CDK2 and CDK1), there are many other CDKs, which perform a function in cell progression, e.g., control the transcription or cell differentiation. Transcriptional CDKs (7, 8, 9, 11, 12, 13, 19 and 20) are essential for the process of synthesizing RNA from DNA [2,6,7]. For example, CDK7 in a complex with cyclin H phosphorylates the carboxy-terminal domain of RNA polymerase II [6].

If there are any abnormalities during cell progression, the cycle can be arrested in the checkpoints—mechanisms which control the progression. This is possible due to the CIP/KIP and INK4 families of the proteins called CDK inhibitors (CDKIs). The CIP/KIP family has three members: p27^KIP1^, p57^KIP2^ and p21^CIP1^ (also known as p27, p57 and p21), which are able to inhibit the cell cycle in the G1 phase by binding cyclin–kinase complexes, including cyclin A–CDK2, cyclin E–CDK2 and cyclin D–CDK4/6. In addition to inhibiting proliferation, p21^CIP1^ is also able to perform other functions such as regulating transcription or apoptosis. Proteins p16^INK4A^, p15^INKB^, p18^INK4C^ and p19^INK4D^ (or just p16, p15, p18 and p19), which are INK4 family members, can bind CDK4 and CDK6 kinases and disturb their binding to cyclin D [5,6]. The stages of the cell cycle and the role of the INK4 and CIP/KIP families are presented in the figure below (Figure 1).

## 3. CDK Inhibitors in Cancer Therapy

The first-generation CDK inhibitors, used in the clinical trials, were first extracted from natural sources (e.g., plants or bacteria), and they started to be applied over 30 years ago. They are unselective (also known as pan-CDK inhibitors), which means they can inhibit many CDKs. One of them, flavopiridol, is an inhibitor of CDK 1, 2, 4, 6, 7 and 9. It was first extracted from an Indian plant and is able to arrest the cell cycle in vitro in the G1/S and G2/M checkpoints. Unfortunately, much research showed that flavopiridol causes serious side effects during the clinical studies. Due to its high cytotoxicity and low efficiency, it is not commonly used in monotherapy [6,7,8].

The second-generation pan-CDK inhibitors, including dinaciclib and AT7519, have been more selective and effective compared to the first-generation ones. Much research relevant to the use of second-generation pan inhibitors in synergistic therapies has been conducted, with the purpose of reducing their cytotoxicity. Some of those inhibitors have been already used in phase I, II or even III clinical trials in various types of cancer. Since they still have not been fully selective and the mechanisms of their work have remained partly unknown, scientists developed the third-generation CDKIs, which were a breakthrough in cancer therapy. Palbociclib, abemaciclib and ribociclib selectively target CDK4 and CDK6 kinases and were approved by the Food and Drug Administration (FDA) in the years 2013–2017 for clinical treatment. All of them are being applied in ER+/HER− advanced breast cancer treatment and also play an important role in many synergistic therapies. CDK4/6 selective inhibitors arrest the cell cycle between the G1 and S phases by blocking RB1 phosphorylation but are also able to activate other anticancer mechanisms like enhancing cytostasis, regulating cell metabolism or inducing senescence and immune responses [7,8]. Some research also focuses on the biomechanisms of targeting CDKs with CDKIs. Most inhibitors have traditionally targeted the ATP-binding pockets of kinases, which has proven challenging due to low selectivity. Recent studies are now focusing on targeting hydrophobic pockets outside the ATP-binding site as a more effective therapeutic approach [9].

In melanoma, CDK4/6 inhibitors, especially the oral inhibitor palbociclib, have been applied in combination with other inhibitors, which are targeted at BRAF and MEK. Unfortunately, the effects of these combinations have been quickly declined by drug resistance. It is suggested that analyzing the mechanisms of acquired resistance to CDKIs could lead to the identification of new biomarkers and strategies for more effective anti-cancer therapies [10]. In recent years, much research has focused on solving this issue, which includes searching for new synergistic therapies with the use of CDK4/6 inhibitors [2,7,8].

## 4. Dysregulation of the Cell Cycle in Melanoma

Dysregulation in the components of the RB pathway (p16–cyclin D–CDK4/6–RB) are responsible for 90% of melanoma cases [5]. They occur in familial melanomas more frequently than in non-hereditary cases of that cancer [1]. Common aberrations, which have influence on melanoma development, include CDKN2A (p16 coding gene) mutations and deletions as well as CDK4/6 genes and CCND1 (cyclin D1 coding gene) amplification. The sequential loss of p16, caused by CDKN2A deletion, is considered the main reason for unregulated cell proliferation in melanoma [2,5] (Figure 2). Mutations that occur in the p16 inhibitor prevent its binding to CDK4 kinase, which causes the fact that p16 cannot stop the cell from entering the S phase of the cycle [1].

However, the mechanisms of cell cycle dysregulation are often more complicated and barely discovered in rare subtypes of melanoma. CDK4 or p16 aberrations often correlate with dysregulations of other molecular pathways, which suggest considering synergistic therapies. For example, in mucosal melanoma, there is a frequent occurrence of CDK4 protein expression in combination with TERT (telomerase reverse transcriptase) amplification [11]. There is a similar situation in many acral lentiginous melanoma cases, where gains of CDK4/CCND1 or loss of CDKN2A, as well as TERT aberrations, have been commonly observed [12]. There is also a growing interest in the interaction between RB and RAS/RAF/MEK/ERK pathways. Moreover, many researchers suggest that targeting the RB pathway in synergistic therapies may overcome resistance to BRAF and MEK inhibitors in melanoma [5].

Unfortunately, in many cases, rare forms of melanoma, such as mucosal melanoma, are treated with the same therapies as regular cutaneous melanoma, which does not bring the expected results [11]. Gene sequencing technology, which has developed in recent years, brought us a clearer view of the molecular landscape and potential oncogenic drivers [2].

Treatment targeted at CDK4/6 and other cyclin-dependent kinases, as well as combination therapies, constitute common subjects of current research. In the following sections of this article, we present preclinical (Table 1) and clinical (Table 2) studies concerned around this kind of therapy in rare forms of melanoma.

**Table 1 molecules-29-05239-t001:** A list of the clinical studies related to the use of CDK inhibitors in the rare types of melanoma.

Type of CDKI	CDKI	Target	Design	Type of the Preclinical Trial	Type of Melanoma	Reference
pan-CDKI	Flavopiridol (**1**—Figure 3)	CDK1, CDK2, CDK4, CDK6, CDK7, CDK9	Flavopiridol + quisinostat (HDAC inhibitor)	In vitro cell lines	Uveal metastatic melanoma	Heijkants et al. (2018) [13]
pan-CDKI	P1446A-05 (**2**—Figure 3)	CDK1, CDK4, CDK9	P1446A-05	In vitro cell lines	Uveal melanoma	Eliades et al. (2016) [14]
pan-CDKI	AT7519 (**3**—Figure 3)	CDK1, CDK2, CDK4, CDK5, CDK9	AT7519	In vivo PDX trial	Mucosal melanoma	Xu et al. (2019) [15]
pan-CDKI	AT7519 (**3**—Figure 3)	CDK1, CDK2, CDK4, CDK5, CDK9	AT7519	In vitro cell lines + in vivo PDX trial	Acral melanoma	Kong et al. (2017) [16]
Selective CDKI	Palbociclib (**4**—Figure 3)	CDK4, CDK6	Palbociclib	In vivo PDX trial	Mucosal melanoma	Zhou et al. (2019) [17]
Selective CDKI	Palbociclib (**4**—Figure 3)	CDK4, CDK6	Palbociclib	In vivo PDX trial	Mucosal melanoma	Xu et al. (2019) [15]
Selective CDKI	Palbociclib (**4**—Figure 3)	CDK4, CDK6	Palbociclib + MEKI (trametinib and PD0325901); palbociclib + MEKI + IACS-010759 (OxPhos inhibitor)	In vitro cell lines + In vivo PDX trial	Uveal melanoma	Teh et al. (2020) [18]
Selective CDKI	Palbociclib (**4**—Figure 3)	CDK4, CDK6	Palbociclib	In vitro cell lines + in vivo PDX trial	Acral melanoma	Kong et al. (2017) [16]
Selective CDKI	Palbociclib (**4**—Figure 3)	CDK4, CDK6	Palbociclib + ERK/MEK inhibitors	In vitro cell lines + in vivo PDX trial	Acral lentiginous melanoma	Jagirdar et al. (2024) [19]
Selective CDKI	Abemaciclib (**5**—Figure 3)	CDK4, CDK6	Abemaciclib + dacarbazine	In vitro cell lines	Head and neck mucosal melanoma	Lyu et al. (2021) [20]
Selective CDKI	Abemaciclib (**5**—Figure 3)	CDK4, CDK6	Abemaciclib; abemaciclib + cMET inhibitor	In vitro cell lines	Uveal melanoma with liver metastases	Ohara et al. (2021) [21]
Selective CDKI	Abemaciclib (**5**—Figure 3)	CDK4, CDK6	Abemaciclib	In vitro cell lines + in vivo PDX trial	Acral melanoma	Kong et al. (2017) [16]
Selective CDKI	Abemaciclib (**5**—Figure 3)	CDK4, CDK6	Abemaciclib + ERK/MEK inhibitors	In vitro cell lines + in vivo PDX trial	Acral lentiginous melanoma	Jagirdar et al. (2024) [19]
Selective CDKI	Dalpiciclib (**6**—Figure 3)	CDK4, CDK6	Dalpiciclib	In vitro cell lines + in vivo PDX trial	Head and neck mucosal melanoma	Shi et al. (2024) [22]
Selective CDKI	Ribociclib (**7**—Figure 3)	CDK4, CDK6	Ribociclib	In vitro cell lines + in vivo PDX trial	Acral melanoma	Kong et al. (2017) [16]
Selective CDKI	Ribociclib (**7**—Figure 3)	CDK4, CDK6	Ribociclib + ERK/MEK inhibitors	In vitro cell lines + in vivo PDX trial	Acral lentiginous melanoma	Jagirdar et al. (2024) [19]
Selective CDKI	SNS-032 (**8**—Figure 3)	CDK7, CDK9	SNS-032	In vitro cell lines	Uveal melanoma with liver metastases	Zhang et al. (2019) [23]
Selective CDKI	Seliciclib (**9**—Figure 3)	CDK2	Seliciclib	In vitro cell lines + in vivo PDX trial	Acral melanoma with lymph node metastases	Namiki et al. (2015) [24]

**Table 2 molecules-29-05239-t002:** A list of the preclinical studies with CDK-targeting in the rare types of melanoma.

Type of CDKI	CDKI	Target	Design	Phases	Conditions	Reference
1st generation pan-CDKI	Flavopiridol (**1**—Figure 3)	CDK1, CDK2, CDK4, CDK6, CDK7, CDK9	FOLFIRI + flavopiridol	I	Oral mucosal melanoma	Dickson et al. (2010) [25]
2nd generation pan-CDKI	Dinaciclib (**10**—Figure 3)	CDK1, CDK2, CDK5, CDK9	Dinaciclib	II	Metastatic melanoma of mucosal origin	Lao et al. (2024) [26]
Selective CDKI	Palbociclib (**4**—Figure 3)	CDK4, CDK6	Palbociclib	II	Advanced acral melanoma	Mao et al. (2021) [27]
Selective CDKI	Dalpiciclib Dalpiciclib (**6**—Figure 3)	CDK4, CDK6	Dalpiciclib	II	Head and neck mucosal melanoma	Shi et al. (2024) [22]

## 5. Mucosal Melanoma

Mucosal melanoma (MM) constitutes less than 2% of all melanomas [28]. It occurs with about equal frequency in all races (African, Asian, Hispanic, Caucasian), in various mucosal membranes. It develops from melanocytes that originate from the neural crest cells. They are able to migrate within the basal layer of the epithelium in various body regions. MM mostly occurs in the head and neck regions (including the sinonasal and oropharyngeal area) and the gastrointestinal, female genital and urinary tracts [28]. It is often diagnosed in its more advanced stage and is more resistant to immunotherapy treatment than cutaneous melanoma (CM) [11,28,29], which leads to poor prognosis, with about a 15% chance of 5-year survival [11,15,28,30]. In the most advanced stages, the average survival time reaches only a few months [15]. Due to this issue, scientists started looking for biomarkers and metabolic pathways, characteristic for MM, to define new therapeutic strategies for treating this cancer. The fact that CDK4 amplification was frequently observed in MM made scientists discover the role of CDKs and CDKIs in this type of cancer [30,31,32,33].

Dickson et al. (2010) conducted a phase I trial of FOLFIRI therapy combined with flavopiridol in patients with advanced solid tumors (Table 2; Figure 4) [25]. FLOFIRI is a combination of irinotecan, fluorouracil and leucovorin commonly used in colon cancer treatment. One of the patients had a history of oral mucosal melanoma that had been surgically removed. Since the metastases to lymph nodes were confirmed, they started a FOLFIRI + flavopiridol therapy, which took 10 months for a complete response achievement. Six months later, the hypermetabolic lymph node had been noticed, but after the resection, no sign of recurrence of melanoma was detected. This case suggested that CDK inhibitors in combination with FOLFIRI therapy may be a potential strategy for mucosal melanoma treatment. Further investigation by Kim et al. (2017) highlighted the potential of combining an MEK inhibitor with CDK4/6 inhibitors for treating MM with oncogenic BRAF fusions [34]. Subsequent studies described the extensive mutational profile of oral MM, exposing CDK4 and TERT (telomerase reverse transcriptase) common amplifications [35]. CDK4 amplifications frequently co-occur with amplifications in TERT or MDM2 (mouse double minute 2 homolog), emphasizing their potential relevance in MM pathogenesis [28,36,37].

The analysis by Zhou et al. (2019) featured a whole-genome sequencing (WGS) of the MM samples, finding amplifications of CDK4 in over 50% of the cases examined [17]. Selective CDK4/6 inhibitor palbociclib was later tested in a preclinical trial with an employment of patient-derived xenograft (PDX) mouse models. Xenografts treated with palbociclib were found to harbor decreased phosphorylation of RB protein (Table 1, Figure 4). Another large WGS analysis by Newell et al. (2019) identified a number of MM tumors as potentially sensitive for CDK4/6 and MEK inhibitors [29]. The study involving 213 MM tumor samples taken from Chinese patients indicated that CDK4 inhibitors may arrest tumor proliferation through various signaling pathways [15]. Both palbociclib and a broad-spectrum pan-CDK inhibitor AT7519 had a significant impact on MM cell lines and PDX tumors harboring abnormalities in CDK4 pathway (Table 2, Figure 4). Genomic analysis revealed an array of abnormal copies of cell cycle-related genes, including CDK4, CCND1 and CDKN2A, in MM samples.

A preclinical trial performed by Lyu et al. (2021) showed that amplifications of CDK4 and TERT coding genes occur in MM more frequently in oral regions than in the nasal cavity and sinuses [20]. The researchers pointed out that a knockdown of the CDK4 gene delays G1/S transition in the oral mucosal melanoma cell line. Furthermore, CDK4/6 inhibitor abemaciclib successfully inhibited oral MM cells’ proliferation, synergistically with dacarbazine, a cytostatic drug commonly used for treating metastatic melanoma (Table 1, Figure 4). In a recent study, Shi and colleagues assessed the efficiency of another CDK4/6 inhibitor dalpiciclib in head and neck mucosal melanoma (HNMM) patients with CDK4 amplification using PDX mouse models and patient-derived tumor cells (PDCs) (Table 2) [22]. Dalpiciclib, like most selective CDK4/6 inhibitors, was used before in treating various cancers, especially breast cancer. In this research, it significantly suppressed the growth of tumors with CDK4 amplification in vivo and in vitro but showed a comparatively weak action in xenografts and cells with CDK4 wild type. After the preclinical studies, the researchers treated patients with dalpiciclib to estimate safety and efficiency of this kind of therapy. It was the first phase II clinical trial that showed the high efficiency of CDK4/6 inhibitor in MM and also proved its safety, as there were no serious side effects noted (Table 2, Figure 4). A recent study by Lao et al. (2024) reports on a phase II clinical trial involving patients with metastatic melanoma of cutaneous or mucosal origin, treated with the second-generation pan-CDK inhibitor dinaciclib [26]. However, the research discourages the use of dinaciclib as a standalone therapy due to its high toxicity and low efficacy.

## 6. Uveal Melanoma

Uveal melanoma (UM) accounts for only 5% of all melanomas [38] but is potentially life-threatening and constitutes the most common primary intraocular malignant tumor diagnosed in adult patients. There are three types of UM: choroidal melanoma, which represents 90% of all UM cases; ciliary body melanoma (6% of all UM cases); and iris melanoma (4% of all UM cases). UM develops from melanocytes located in highly pigmented uveal tissue and is more common in older patients of Caucasian race from Northern Europe (eight cases per million persons per year) [38,39]. The 5-year survival rate varies between 76% and 82% but is variable due to metastases. Although some patients, especially with iris melanoma, may notice characteristic symptoms like a change of iris color, there is still no gold standard for the UM prognostication [39]. Scientists continue to search for new potential targeted therapies to find strategies for UM treatment.

Older studies report that p21, p16, p14^ARF^ and CDK4 coding genes are not responsible for genetic susceptibility to UM [40,41,42,43]. However, many other researchers reached the opposite conclusion [44,45,46,47,48,49,50,51]. Inhibiting choroidal melanoma cells with a phosphatidylinositol 3-kinase inhibitor LY294002, performed by Casagrande et al. (1998), caused the cell cycle arrest correlated with CDK4 and CDK2 inhibition and impairment of RB phosphorylation. The researchers claimed that the inhibition of CDK2 was partly caused by the up-regulation of p27 [44]. Expression of p27 was also noticed in subsequent research in human choroidal melanoma tumor cells (OCM-1), with a high level of MARCKS (myristoylated alanine-rich C kinase substrate) protein [45]. A comparison of normal and transformed human choroidal melanocytes performed by Mouriaux et al. (1998 and 1999) showed that p16, p21 and p27 coding genes are implicated in the progression of choroidal melanoma tumors [46,47]. In another study, an MEK inhibitor combined with the DNA methyltransferase inhibitor increased the expression of p21 in UM [52].

A significant activity of a novel pan-CDK inhibitor P1446A-05, against both cutaneous and uveal melanoma, was demonstrated by Eliades et al. (2016) on cell lines by arresting the cell cycle in the G2/M phase (Table 1, Figure 4) [14]. Further research proved the synergistic impact of pan-CDK inhibitor flavopiridol and histone deacetylase (HDAC) inhibitor quisinostat on uveal metastatic melanoma cell lines [13] (Table 1, Figure 4). Flavopiridol inhibited cell proliferation at distinct stages of the cycle and, in combination with quisinostat, inducted apoptosis, which led to UM cell death. A study by Zhang et al. (2019 and 2021) indicated the potential of a selective CDK7/9 inhibitor SNS-032 in treating UM with liver metastases. The researchers suggested that the elevated activity of transcription of oncogenes may cause liver metastases in UM and that inhibition of this transcription could reduce the metastases by abrogating the oncogenes [23]. Indeed, SNS-032 significantly arrested cell proliferation and inducted apoptosis in human UM cell lines (Table 1, Figure 4). It also suppressed tumor growth in PDX mouse models, reduced invasive phenotypes of UM cells and eradicated cancer stem-like cells (CSCs). Eventually, SNS-032 repressed cell motility and liver metastases, which suggest it as a promising therapy option for metastatic UM.

As dysregulations in the RB pathway are frequently observed in UM (about 90% of all cases), recent studies focus on targeting CDK4 and CDK6 in this melanoma subtype [21]. A synergistic effect of palbociclib and a MEK inhibitor was investigated by Teh et al. in 2020 [18]. The combination decreased the expression of cell cycle proteins in vitro but did not induct apoptosis. In vivo, palbociclib arrested tumor growth with almost the same effectivity when combined with MEKI. As the inhibitors did not cause a full tumor regression, the researchers performed RNA sequencing, which revealed upregulation of the oxidative phosphorylation (OxPhos) pathway in UM resistant to MEKI and tolerant of the CDK4/6 inhibitor. The combination of MEKI and palbociclib with OxPhos inhibitor IACS-010759 led to apoptosis and tumor regression (Table 1, Figure 4). An antiproliferative and suppressing efficiency of abemaciclib was indicated by Ohara et al. in 2021 on metastatic UM cell lines (Table 1, Figure 4) [21]. However, the effect was significantly decreased by a hepatocyte growth factor (HGF), which possibly was also responsible for liver metastases. The researchers combined abemaciclib with an inhibitor directed at the cMET–tyrosine kinase receptor activated by HGF and tested the synergistic activity on human HGF knock-in xenograft mouse models. This combination significantly suppressed the tumor growth.

In the latest study, Onken et al. (2023) identified mutant constitutively active G protein α-subunits Gq and G11 as UM tumor drivers. Inhibiting them with a specific Gq/11 inhibitor caused CDK1 and CDK2 deactivation [53], highlighting their role in cell progression in uveal melanoma.

## 7. Acral Melanoma

Acral melanoma (AM) constitutes 2–3% of all melanoma cases and often occurs in darker-skinned patients. Although AM is very rare, it is considered the most common melanoma subtype in African, Chinese, Korean, Singaporean and Hispanic populations [54]. In some Asian or Latin American countries, it can even exceed 50%. In South Africa, AM reaches 65% of all melanomas in Black individuals [55]. It arises in the acral sites, primarily on the soles of the feet, but also on the palms of the hands and in the nail beds [2,12,54,55]. The term “acral melanoma” is often used interchangeably with “acral lentiginous melanoma (ALM)”. However, Basurto-Lozada et al. (2021) warn us that the nodular or superficial spreading melanoma subtypes, especially on the dorsum of the hands or feet, cannot be classified as ALM [12]. AM is often diagnosed in its late stages and mistaken with other dermatological afflictions. Patients have worse 5- and 10-year survival rates than in other cutaneous melanoma subtypes. AM is characterized with embryonic origin of melanocytes and a large amount of chromosomal aberrations [12,55]. Although the mutational landscape in AM is very different and still not fully discovered, much research highlights dysregulations in the CDK4 pathway [12,16,19,24,27,30,56,57,58,59,60,61,62,63,64,65,66,67,68,69,70]. Common amplification of the CDK4 coding gene was emphasized by Curtin et al. (2014) in both acral and mucosal melanoma [55]. However, other research described Caucasian AM patients with no CDKN2A/CDK4 gene mutations [67].

Namiki et al. (2015) studied the way in which NUAK2 and PI3K pathways control the expression of CDK2 in the AM cell line [24]. They observed that those pathways may increase cancer cell proliferation in the S phase and that CDK2 is a potential target in NUAK2-amplified AM cells. Then, they analyzed the efficacy of a selective CDK1/2 inhibitor seliciclib (mentioned as roscovitine) in vitro and in vivo. Seliciclib significantly inhibited NUAK2-amplified cell proliferation in vitro and tumor growth in mice, which suggests using CDK2 inhibitors as a promising option for AM treatment (Table 1, Figure 4). Another analysis of 514 AM samples, demonstrated by Kong et al. in 2017, focused on CDK4 pathway aberrations and the efficiency of CDK4/6 inhibitors [16]. In total, aberrations in CDK4, CCND1 and CDKN2A genes were observed in 82,7% of the samples. The pan-CDK inhibitor AT7519 and selective CDK4/6 inhibitors palbociclib, ribociclib and abemaciclib were tested on AM cell lines and PDX models for the purpose of clarifying the utility of CDKIs in AM treatment. In vitro, the pan-CDK inhibitor turned out to be more effective than the selective ones. Surprisingly, the selective inhibitors did not bring equal effects. In vivo, AT7519 effectively inhibited the AM tumor growth bearing CDK4 gain + CCND1 gain, CDK4 gain + p16^INK4A^ loss or only CDK4 gain, whereas palbociclib was effective in mouse models with CDK4 gain + CCND1 gain and CDK4 gain + p16^INK4A^ loss (Table 1, Figure 4). Thanks to this research, CDK4/6 inhibitors may be a good therapy option for AM patients with concurrent two aberrations of CDK4 gain, p16^INK4A^ loss or CCND1 gain.

Comparison of acral melanoma and subungual melanoma (SUM), which is often included as a subtype of AM, has been provided by Holman et al. in 2020 [63]. They discovered that CDK4/CCND1 amplifications were more frequent in SUM, whereas CDKN2A/B loss occurred more often in AM. SUM was common in younger patients with frequently reported melanoma on the hand and injuries in the tumor area. The researchers suggest using CDK4/6 as well as KIT and mTOR inhibitors in the treatment of SUM. The fact that CDK4/CCND1 amplifications are more common in SUM than in different subtypes of AM was noticed 3 years earlier by Haugh et al. in 2018 [68]. However, the efficiency and safety of this kind of therapy have not been analyzed yet, so it must be tested in future trials [63,64]. Another whole-genome sequencing provided a clearer genomic landscape of AM among East Asian patients [65]. Thirteen patients with melanoma from Taiwan and Singapore (including eight with AM) were examined. Recurrent copy number gains or amplifications of CCND1 and CDK4 genes as well as deletions in CDKN2A/CDKN2B were observed most frequently in AM samples.

Mao et al. (2021) performed a phase II clinical trial in which 15 patients with advanced AM with CDK4 pathway aberrations were treated with oral palbociclib [27,66]. They analyzed whole-exome sequencing and multiplex fluorescence immunohistochemistry (IHC) samples of nine patients. The CDK4/6 inhibitor was preliminarily effective, especially in patients with amplification or high protein levels of minichromosome maintenance protein 7 (MCM7) (Table 2, Figure 4). An acceptable safety profile was stated, and side effects of grade I or II (neutropenia/leukopenia and anemia) were noted, but there were no serious adverse events caused by palbociclib. The researchers demonstrated that CDK4/6 inhibitor treatment may be a good option for patients with simultaneous CCND1 gain, CDK4 gain and CDKN2A loss. They also suggested that the MCM7 status and the JAK-STAT pathway could be predictive biomarkers of CDK4/6 inhibitors in AM.

Sometimes, acral lentiginous melanoma cells acquire resistance to CDK4/6 inhibitors [19]. As one of the reasons for that issue, Jagirdar et al. (2024) reported hyperactivation of the mitogen-activated protein kinase (MAPK) pathway and elevated cyclin D1 [19]. The research included inhibiting ALM cells and PDX mouse models with CDK4/6 inhibitors combined with ERK/MEK inhibitors or with silencing of cyclin D1 (Table 1, Figure 4). These combinations effectively reserved CDK4/6 inhibitor resistance in vitro and in vivo. The analysis proved the saliency of the MAPK pathway in mechanisms of CDK4/6 inhibitor resistance in ALM cells.

## 8. Desmoplastic and Amelanotic Melanoma

Desmoplastic melanoma (DM) is a very rare variant of melanoma frequently found in elderly male individuals on the head and neck cutis exposed on UV radiation [71]. This form of melanoma is extremely hard to diagnose in its early stages, as it is often flesh-colored and the pigment is rarely visible. Histologically, it is easy to confuse it with non-melanocytic spindle cell tumors or other melanotic skin diseases, e.g., desmoplastic Spitz nevi [72,73]. In a study by Hilliard et al. (2009), a distinct response for the p16 antibody in Spitz nevi was observed, where DM samples were mostly barely stained or not stained at all [73]. Kiuru et al. (2012) reached similar conclusions [74]. However, in research by Blokhin et al. (2013), the results for 22 DM tumors were not identical [75]. In some of them, a diffuse staining was noticed, and some even gave positive response. Further investigation reported p16 expression in all 40 DM samples [76]. Other research showed that p53 expression corelated with Ki-67 and male gender in patients with DM [77]. Although since that time some other studies have been performed [78,79,80], the role of p16 in desmoplastic melanoma is still unclear, and it certainly needs more deep scientific research to be conducted.

In total, 2–10% of all melanoma cases are devoid of melanin and have nearly total lack of visual pigment [81]. To our knowledge, research relevant to amelanotic melanoma is very sparing with the role of any cell cycle-controlling proteins. Ghiorzo et al. (2009) observed a rare p14^ARF^ splice germline mutation in an Italian family with amelanotic melanoma [82]. In the amelanotic melanoma cell line (Ab), p21^CIP1/WAF1^ and p27^KIP1^ were upregulated following treatment with proteasome inhibitors; however, this did not lead to cell cycle arrest. [83]. It is essential to conduct more research connected to CDK inhibitors in desmoplastic and amelanotic melanoma.

## 9. Conclusions

Cell cycle dysregulations, particularly in the RB pathway, are commonly observed across various forms of melanoma. The combined inhibition of CDK4, MDM2 and TERT presents a promising therapeutic strategy for mucosal melanoma, as amplifications of these three genes frequently co-occur in this cancer subtype. In recent years, CDK4/6 inhibitors such as palbociclib, abemaciclib and dalpiciclib have been successfully tested on mucosal melanoma cell lines and xenograft mouse models. Additionally, patients with head and neck mucosal melanoma were treated with dalpiciclib, demonstrating both efficacy and safety, with no serious side effects reported.

In earlier studies, the role of CDKs and CDKIs in uveal melanoma was often questioned, as amplifications were not consistently observed. However, the synergistic effect of flavopiridol and an HDAC inhibitor, along with the anti-proliferative and pro-apoptotic action of a CDK7/9 inhibitor, was later demonstrated in uveal melanoma cell lines. Additionally, the CDK7/9 inhibitor was shown to inhibit UM tumor growth and liver metastases, highlighting the need for further research on this therapeutic approach. Palbociclib, when combined with MEK and OxPhos inhibitors, exhibited tumor-suppressing efficacy, and in another study, abemaciclib and a hepatocyte growth factor inhibitor synergistically inhibited tumor growth in xenografts. Notably, to our knowledge, no previous reviews have provided a comprehensive summary of preclinical trials related to the use of CDKIs in uveal melanoma therapy.

Acral melanoma (AM), commonly found in the glabrous skin, is extremely rare in the European population but more prevalent among Hispanic, Asian and African patients. CDK2 inhibitors have shown promise as a therapeutic option for AM cases with NUAK2 expression. Multiple studies confirm that both pan-CDK and selective CDK4/6 inhibitors could represent potential treatment strategies for AM. A recent study demonstrated the preliminary efficacy and safety of palbociclib in AM treatment, suggesting that MCM7 and the JAK-STAT pathway could serve as potential biomarkers for CDK4/6 inhibitors in AM. Another study revealed the synergistic efficacy of combining CDK4/6 inhibitors with ERK/MEK inhibitors.

Desmoplastic and amelanotic melanomas are exceedingly rare, and the specific cell cycle aberrations driving these cancers remain largely unidentified. However, several studies on these forms could provide valuable insights. Current research consistently highlights CDKIs as a promising therapeutic strategy for treating rare melanoma subtypes.

## 10. Future Perspectives

Enhancing our understanding of CDK inhibitors and cell cycle mechanisms is crucial for effective cancer treatment, particularly the role of CDK4, as it is essential for driving the transition from the G1 to S phases of the cell cycle, a critical step in cell proliferation and survival. In cancer, cell cycle dysfunction leads to uncontrolled cell division, making CDKs critical targets for developing new therapeutic approaches. A deeper exploration of the mechanisms behind cell cycle abnormalities in melanoma has the potential to identify novel diagnostic markers that could improve early detection and treatment options. Identifying these biomarkers would not only enhance the ability to predict disease progression but also offer personalized treatment strategies, potentially improving patient outcomes. Furthermore, advancements in this area could support the development of more selective CDK inhibitors, reducing safety risks and more effectively addressing resistance challenges seen in clinical practice.

Despite considerable progress in preclinical research, the reliance on cell lines and xenograft models fails to fully capture the complexity of human tumors. This underscores the necessity for further clinical research to validate findings in diverse patient populations. By advancing clinical trials, we can gain vital insights into the efficacy and safety of CDK inhibitors, paving the way for more effective treatments that could become integral components of standard melanoma therapy.

## Figures and Tables

**Figure 1 molecules-29-05239-f001:**
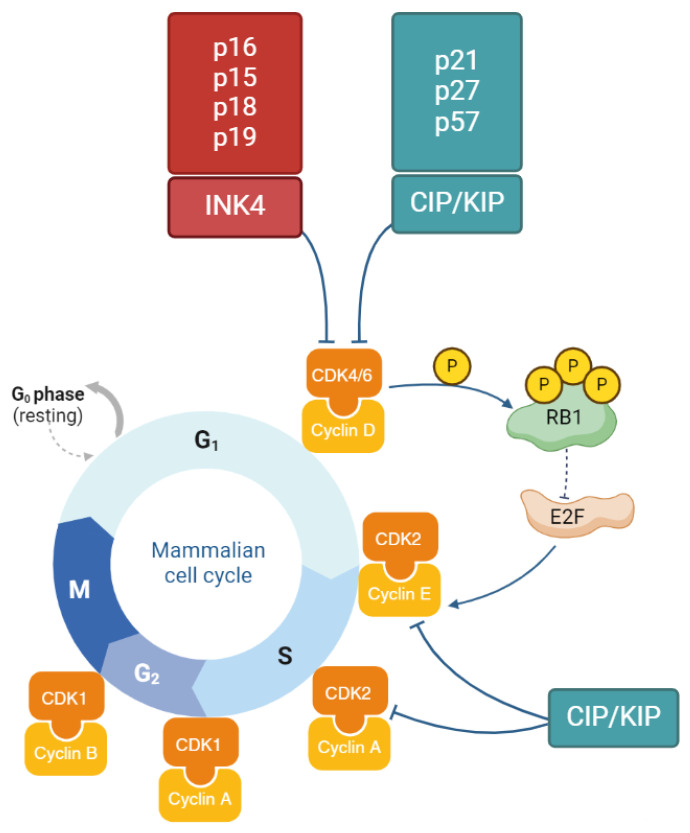
Regulation of the mammalian cell cycle and the role of the INK4 and CIP/KIP families in inhibiting cyclin–kinase complexes. The cell cycle consists of four main phases (G1, S, G2, M) and a resting phase, G0. The cycle is regulated by cyclin–CDK complexes. During the G1 phase, CDK4/6 binds with cyclin D and phosphorylates the retinoblastoma protein (RB1), leading to the release of E2F from the RB1-E2F complex. E2F then initiates the synthesis of cyclin E, which subsequently binds with CDK2, allowing the cell to enter the S phase. The later phases are regulated by the cyclin A–CDK2, cyclin A–CDK1, and cyclin B–CDK1 complexes. If abnormalities occur during cell cycle progression, the cycle can be halted by two families of CDK inhibitors (CDKIs): the INK4 family (p16, p15, p18, p19), which inhibits the cyclin D–CDK4/6 complex, and the CIP/KIP family (p21, p27, p57), which can inhibit CDK2 from binding to cyclin E or cyclin A, as well as CDK4/6 from binding to cyclin D. (Illustration created with BioRender.com (accessed on 2 November 2024).

**Figure 2 molecules-29-05239-f002:**
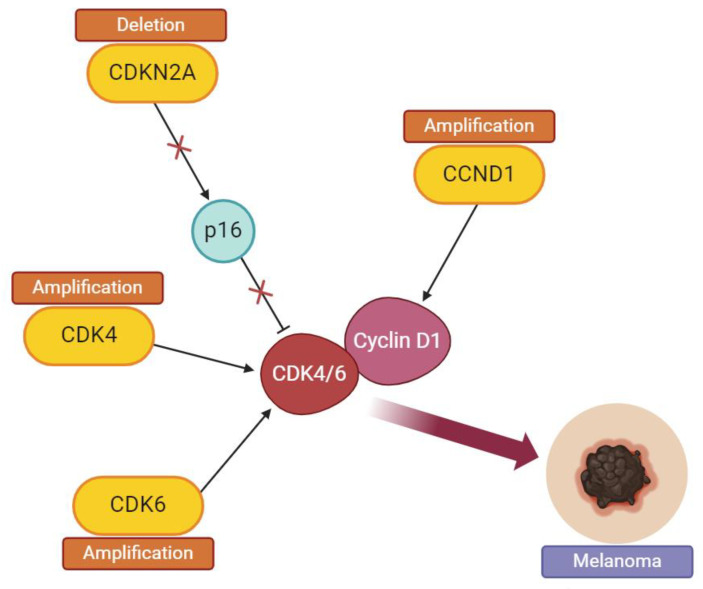
Deletion of CDKN2A as well as amplification of CCND1 or CDK4/6 coding genes frequently constitute the reason for an uncontrollable cell proliferation in melanoma. Deletion of CDKN2A (p16 coding gene) prevents the p16 inhibitor from binding with CDK4/6, which leads to uncontrolled progression of the cell. Amplifications of CDK4, CDK6 and cyclin D1 (CCND1) coding genes are also common aberrations that lead to melanoma. (Illustration created with BioRender.com. URL (accessed on 2 November 2024)).

**Figure 3 molecules-29-05239-f003:**
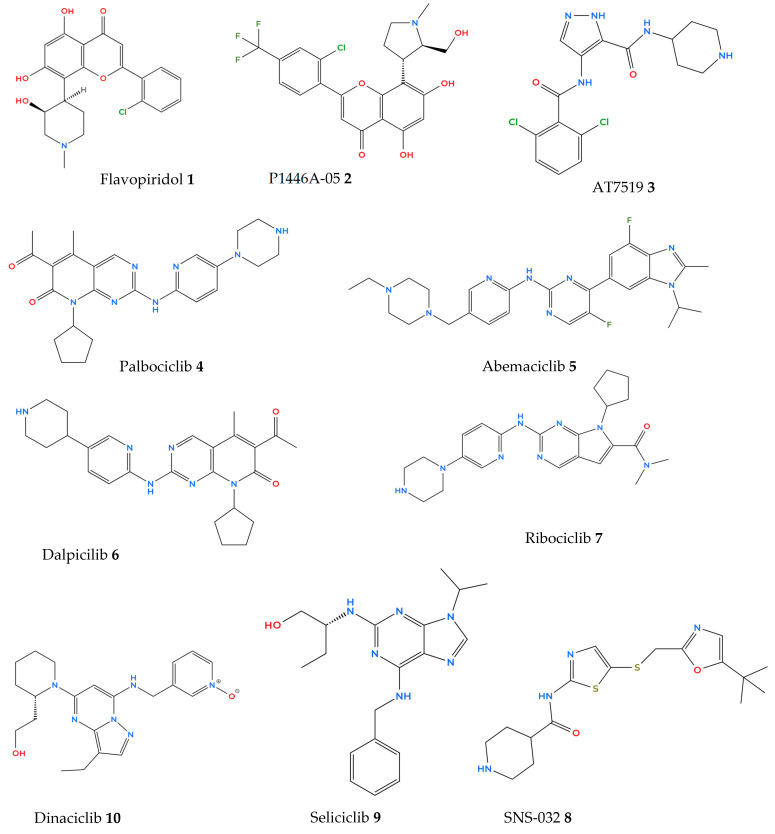
Chemical structures of CDKIs used in rare types of melanoma therapies. (Illustration created with MolView URL (accessed on 2 November 2024)).

**Figure 4 molecules-29-05239-f004:**
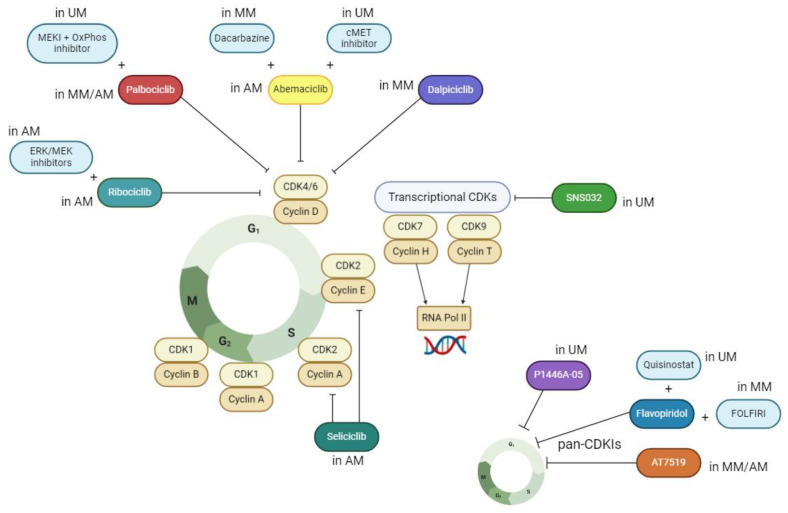
A summary of preclinical and clinical research connected to the use of CDKIs in treatment of mucosal melanoma (MM), uveal melanoma (UM) and acral melanoma (AM). Several studies pointed out an inhibiting action of selective CDK4/6 inhibitors (palbociclib, abemaciclib, ribociclib, dalpiciclib), broad-spectrum pan-CDK inhibitors (P1446A-05, flavopiridol, AT7519), CDK2 inhibitor seliciclib and CDK7/9 inhibitor SNS032 in single or synergistic therapies in various forms of melanoma (MM, UM, AM) treatment. (Illustration created with BioRender.com URL (accessed on 2 November 2024)).

## Data Availability

No new data were created or analyzed in this study. Data sharing is not applicable to this article.

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
