# Peer review of "Cyclin-Dependent Kinase Inhibitors in the Rare Subtypes of Melanoma Therapy"

_molecules, 2024, doi:10.3390/molecules29225239_

Round 1

Reviewer 1 Report

Comments and Suggestions for Authors

The authors have presented a review article regarding the function of CDK inhibitors in melanomas. The authors examined the rationale for these inhibitors as prospective therapeutic options for the treatment of breast cancer, as licensed by the FDA. The authors effectively emphasized the historical and contemporary studies pertinent to the targeting of various CDKs in mucosal, uveal, and acral melanomas. The study additionally addressed the amelanotic and desmoplastic variants of melanoma and emphasized the necessity for future research to elucidate the cell cycle dysregulations that contribute to cancer proliferation. The article is supported by 3 very attractive Figures as well 2 very informative table.  The bibliography is up to date.

There are some minor issue as following.

1.      The Figure legend should be expanded to make it more informative for readers.

2.      On page 7-line 172The statement “Mucosal melanoma (MM) constitutes less than 2% of all melanomas” should be provided with references.

3.      On page 7 and line 185  “ Dickson et al. ( reference)---------. Similarly  line 1 Kim et al (reference )-------.

4.      Conclusion and future prospective should be separated and future prospective should  contain what strategy should be adopted to accelerate research in this area.

Comments on the Quality of English Language

The authors have presented a review article regarding the function of CDK inhibitors in melanomas. The authors examined the rationale for these inhibitors as prospective therapeutic options for the treatment of breast cancer, as licensed by the FDA. The authors effectively emphasized the historical and contemporary studies pertinent to the targeting of various CDKs in mucosal, uveal, and acral melanomas. The study additionally addressed the amelanotic and desmoplastic variants of melanoma and emphasized the necessity for future research to elucidate the cell cycle dysregulations that contribute to cancer proliferation. The article is supported by 3 very attractive Figures as well 2 very informative table.  The bibliography is up to date.

There are some minor issue as following.

1.      The Figure legend should be expanded to make it more informative for readers.

2.      On page 7-line 172The statement “Mucosal melanoma (MM) constitutes less than 2% of all melanomas” should be provided with references.

3.      On page 7 and line 185  “ Dickson et al. ( reference)---------. Similarly  line 1 Kim et al (reference )-------.

4.      Conclusion and future prospective should be separated and future prospective should  contain what strategy should be adopted to accelerate research in this area.

Author Response

Comment 1:
The Figure legend should be expanded to make it more informative for readers.
Response 1:
Thank you for pointing this out. We have expanded the descriptions of all three figures, making them more specific and detailed (Page 3, line 110; Page 5, line 165; and Page 12, line 372).

Comment 2:
On page 7-line 172 the statement “Mucosal melanoma (MM) constitutes less than 2% of all melanomas” should be provided with references.
Response 2:
Thank you for the comment. We have added the reference immediately following the sentence (Page 8, line 197).

Comment 3:
On page 7 and line 185  “ Dickson et al. ( reference)---------. Similarly  line 1 Kim et al (reference )-------.
Response 3:
Thank you. We have adjusted the references throughout the paper, placing them immediately after the authors' names instead of at the end of the sentence.

Comment 4:
Conclusion and future prospective should be separated and future prospective should  contain what strategy should be adopted to accelerate research in this area.
Response 4:
Thank you for the suggestion. We have separated the “Future perspectives” section from the “Conclusion” subsection. Additionally, we expanded the future prospects section to include more detailed strategies that should be implemented to advance research on the role of CDK inhibitors in melanoma treatment (Page 14, line 452).

Reviewer 2 Report

Comments and Suggestions for Authors

In this review, the authors focus on the study of cyclin-dependent kinase (CDK) inhibitors in melanoma types with lower incidence, such as mucosal, uveal, acral, amelanotic and desmoplastic melanomas. For this purpose, they displayed an organized structure where they first presented the role of CDK inhibitors in the cell cycle and in cancer. Then, the importance of dysregulation of cell cycle in melanoma, and in each of the rare melanomas, is presented. Finally, the authors propose multitherapy of CDK inhibitors with oxidative phosphorylation and mitogen-activated protein kinase (MEK) inhibitors as a promising strategy against these rare melanoma subtypes. After carefully reading the full text, I consider this manuscript acceptable for publication after appropriate revisions:  

  1. In section 3 “CDK inhibitors in cancer therapy”, authors are recommended to review the articles (O'Leary et al., 2016, PMID: 27030077; Lukasik et al., 2021, PMID: 33802080), where the first one is a highly cited paper of special relevance in the field, while the second one compiles the main mechanisms of action of CDK inhibitors.
  2. In Table 1, “Study/Authors” column, it is difficult to match the citation with the correct reference. For this, you can choose to add the year to each citation (e.g. Heijkants et al., 2018) or you can replace all citations with numbers in the proper order (e.g. [43]). In addition, the font used in Table 1 should be the same as that of the text.
  3. Because each figure should be able to explain itself, it is recommended to add a short description after the title of each figure. This applies to the 3 Figures presented.
  4. In general, English used was correct. There are some minor suggestions and corrections to improve/polish English grammar:
  • In line 60, “…, selectively targeted targeting at specific CDKs,…”
  • In line 79, “which leads to phosphorylating phosphorylation of retinoblastoma…”
  • In line 114, “Many research relevant to the…” check grammar.

  However, the main concern for accepting this work is that there is no evident innovation or relevance compared to what has been reported in the literature (Guo et al., 2020, PMID: 32712436; Garutti et al., 2021, PMID: 34071228), because these works have already reported how some CDK inhibitors act on rare melanomas: Palbociclib (acreal), Abemaciclib (mucosal), SNS032 (uveal). As such, it is possible to observe that the main difference between the previously cited works and this one is the addition of desmoplastic and amelanotic melanoma. Thus, we encourage the authors to emphasize the relevance of this document.

Comments on the Quality of English Language

It is recommended to review the document with the assistance of a native English speaker/teacher to correct grammatical details. There are some minor suggestions and corrections to improve English grammar/language:

·      In line 60, “…, selectively targeted targeting at specific CDKs,…”

·      In line 79, “which leads to phosphorylating phosphorylation of retinoblastoma…”

·      In line 114, “Many research relevant to the…” check grammar.

Author Response

Comment 1:
In section 3 “CDK inhibitors in cancer therapy”, authors are recommended to review the articles (O'Leary et al., 2016, PMID: 27030077; Lukasik et al., 2021, PMID: 33802080), where the first one is a highly cited paper of special relevance in the field, while the second one compiles the main mechanisms of action of CDK inhibitors.
Response 1:
We are very thankful for your recommendations. We took a look at both of the articles, especially on paper by Lukasik et al. and decided to mention them in the references. In the section “CDK inhibitors in cancer therapy” we have added: “Some research also focus on biomechanisms of targeting CDKs with CDKIs. Most of the inhibitors have been often targeted at kinases’ ATP binding pockets, what was very challenging due to the low selectivity. Recent studies focus on targeting hydrophobic pockets outside the ATP binding site as a better therapeutic option[9]” (Page 4 / line 142). We have also mentioned an information taken from article by O'Leary et al.: “In melanoma, CDK4/6 inhibitors, especially oral inhibitor palbociclib, have been applied in combination with other inhibitors, which are targeted at BRAF and MEK. Unfortunately, the effects of these combinations have been quickly declined by drug resistance. However, scientists claim that the if we analyze the mechanisms of acquired resistance to CDKIs, we will be able to identify new biomarkers and strategies for effective anti-cancer therapies[10]. In recent years, many research major on solving this issue, which include searching for new synergistic therapies with the use of CDK4/6 inhibitors[2,7,8]”.

Comment 2:
In Table 1, “Study/Authors” column, it is difficult to match the citation with the correct reference. For this, you can choose to add the year to each citation (e.g. Heijkants et al., 2018) or you can replace all citations with numbers in the proper order (e.g. [43]). In addition, the font used in Table 1 should be the same as that of the text.
Response 2:
We agree with this comment. To ease matching the citation with correct reference, we have added a year in the “Study / authors” column, so now it will surely be more clear. We have also chosen a correct font for both table 1 and table 2 (Page 6 / line 191). 

Comment 3:
Because each figure should be able to explain itself, it is recommended to add a short description after the title of each figure. This applies to the 3 Figures presented.
Response 3:
Thank you for pointing this out. Therefore, we have expanded the descriptions of all three figures, so they are now more specific and contain more details about the figures (Page 3 / line 110; page 5 / line 165 and page 12 / line 372). 

Comment 4:
In general, English used was correct. There are some minor suggestions and corrections to improve/polish English grammar:
In line 60, “…, selectively targeted targeting at specific CDKs,…”
In line 79, “which leads to phosphorylating phosphorylation of retinoblastoma…”
In line 114, “Many research relevant to the…” check grammar.
Response 4:
Thank you for your remarks. We fixed the first two sentences, which you had mentioned and we checked grammar in the third sentence. We have finally changed it to: “Much research relevant to the use of the second generation pan inhibitors in synergistic therapies has been conducted, with the purpose of reducing their cytotoxicity” (Page 4 / line 129).

Comment 5:
However, the main concern for accepting this work is that there is no evident innovation or relevance compared to what has been reported in the literature (Guo et al., 2020, PMID: 32712436; Garutti et al., 2021, PMID: 34071228), because these works have already reported how some CDK inhibitors act on rare melanomas: Palbociclib (acral), Abemaciclib (mucosal), SNS032 (uveal). As such, it is possible to observe that the main difference between the previously cited works and this one is the addition of desmoplastic and amelanotic melanoma. Thus, we encourage the authors to emphasize the relevance of this document.
Response 5:
Thank you for pointing out that issue. As there are many other works concerning on the use of CDK inhibitors in melanoma treatment, we have decided to emphasize the main innovations of our article. Our work summarize the present and older research that focus on CDKIs in uveal melanoma therapy, as this form of melanoma was often not included or only mentioned in previous reviews. To our knowledge, desmoplastic and amelanotic types of melanoma were not mentioned before in articles relevant to CDK inhibitors. To emphasize the relevance of our review, we have described the foregoing facts in the introduction: “We focused especially on the preclinical trials relevant to uveal melanoma, as they were barely mentioned in the previous review articles. To our knowledge, desmoplastic and amelanotic forms of melanoma also weren’t discussed before in terms of the role of CDKs and CDK inhibitors, probably due to a small number of conducted research. That is why we decided to briefly characterize it at the end of the article” (Page 2 / line 72).

Reviewer 3 Report

Comments and Suggestions for Authors

The reviewed article "Cyclin-dependent kinase inhibitors in the rare subtypes of melanoma therapy" raises an interesting topic of the effect of CDK inhibitors in melanoma therapy. The article is very well written, well-prepared figures summarize the topic.

The text is very readable thanks to the division into subsections, the selection of literature is appropriate.

However, I have minor reservations:

- In the introduction, I miss a description of the methodology for preparing a review paper.

- In Table 2, it would be possible to add clinical trials ongoing according to the clinicaltrials database.

Author Response

Comment 1:
In the introduction, I miss a description of the methodology for preparing a review paper.
Response 1:
Thank you for pointing this out. We have mentioned the methodology in the last paragraph of the introduction: “In order to prepare this review, we searched the PubMed database, using the keywords: “uveal melanoma CDK4/6 inhibitors”, “uveal melanoma CDK4 inhibitors”, “uveal melanoma CDK2 inhibitors”, “uveal melanoma CDK7/9 inhibitors” similarly in different types of melanoma and targeted kinases. We focused especially on the preclinical trials relevant to uveal melanoma, as they were barely mentioned in the previous review articles” (Page 2, line 69).

Comment 2:
In Table 2, it would be possible to add clinical trials ongoing according to the clinical trials database.
Response 2:
Thank you for your suggestion. We have searched the database and found one more clinical study (by Lao et al.), that was not mentioned before (Page 7, line 193).

Round 2

Reviewer 2 Report

Comments and Suggestions for Authors

After corrections and suggestions, the document has improved in terms of citing related bibliography, complementing tables and figures and English grammar.

Author Response

Thank you very much